# The Extent of Incorporating Health Education Requirements in Middle School Science Textbooks

**Abdulwali H. Aldahmash** [1,*] and **Sarah A. Almutairi** [2]

1 Curriculum and Instruction Department, College of Education, Excellent Research Centre of Science and Mathematics Education, King Saud University, Riyadh 11362, Saudi Arabia
2 College of Education, King Saud University, Riyadh 11362, Saudi Arabia
* Correspondence: aaldahmash@ksu.edu.sa

**Abstract:** To ensure health sustainability for the next generation, an emphasis should be placed on health prevention and health education. Therefore, it is crucial to educate kids on how to keep themselves healthy in order to promote long-term progress. This study aimed to identify the extent of the inclusion of health education requirements in the content of science textbooks for the intermediate stage of schooling. To fulfill the goal of this study, a validated content-analysis card consisting of twenty-seven indicators categorized into seven requirement themes was used in the analysis of middle-school-level science textbook editions that were taught in the academic year 2022. The results revealed that all health education requirements were inadequately included in the science textbooks, in which the inclusion rate ranged between 5.0% and 1.3%, and the percentage of included health-related themes in the middle school science textbooks, in general, was 2.8%. These findings indicate the necessity for reforming science textbooks for middle school levels so that they include sufficient basic health education requirements to enable students to protect themselves from prevailing diseases across the globe.

**Keywords:** health education requirements; inclusion; science textbooks; intermediate stage; textbooks analysis; pandemics and epidemics

## 1. Introduction

Schools play a significant role in illness prevention and curative domains by assisting individuals in learning about health issues and acquiring the appropriate behaviors toward health risks and diseases [1]. Darling-Hammond and Hyler [2] stressed that the school's role extends beyond preparing students academically, culturally, socially, mentally, and psychologically in a thorough and balanced manner to shaping their ideas of a healthy culture by teaching them the prerequisites for the prevention and successful management of health concerns.

In addition, the nature of the scientific discipline, as the subject most closely related to students' lives and health, should render it a clear role in providing the knowledge and abilities necessary for students to protect themselves from being exposed to dangerous diseases. Science curricula also play a crucial role in helping students develop critical thinking skills and their knowledge of prevailing scientific trends [3]. Moreover, it was asserted [4] that science curricula and literacy are responsible for enabling students to deal with various health problems. This indicates that science curricula play a preventive role in the management of ill-health in students and communities. Providing students with the knowledge required to practice specific healthy behaviors, as well as assisting them in gaining practical experience at home, at school, or in their neighborhoods, will ensure their safety and that of their community.

Modern education emphasizes health and the use of cutting-edge methodologies, as described by Vamos, Okan, Sentell, and Rootman [5]; Henderson, Wolle, Cortese, and McIntosh [6]; and Institute & Medicine, 2003 [7]. Health education literacy can be applied

in several contexts to assist people, their families, and their communities (such as in schools, homes, workplaces, and governments) in achieving their health and wellness objectives. The education domain is a crucial platform for these triumphs since health literacy is a basic process and outcome and can help in accomplishing significant public health goals, such as critical health literacy, as it is focused on supporting successful social and political actions as well as individual activities [5].

Health education literacy may lead to the promotion and maintenance of good health. Consequently, people must acquire and apply health-related information effectively. When literacy is used in a health context, it is referred to as "health literacy", even with regard to facilitating the understanding and communication of health information and concerns [8]. Science curricula should foster health education as a requirement to protect societies from contracting era-specific diseases and to raise community awareness of health issues in a way that will ensure the prevention of diseases, especially in light of the emergence of new health crises. In this context, Bavel, Baicker, and Boggio [9] emphasized that health-promotion and illness-prevention practices are not intended to be used in response to incidents; rather, their importance increases in the face of crises and disasters, such as those caused by pandemics and epidemics, including SARS, avian flu, Ebola, and dengue fever.

Focusing on health prevention and health education is one of the strategies used to support concepts of sustainability in health. Thus, students must be taught how to maintain their health in order to ensure sustainable development for the future generation. Today's schools must bring students out of the shadows of their ignorance and correct their misperceptions of the world around them. It must guide them towards smarter behavior that promotes healthy and sustainable development for both themselves and the next generation. Students' abilities to live long and healthy lives can be ensured through science curriculum reform. Accordingly, healthy and sustainable development can be achieved by integrating current values and standards of health education into science textbooks.

Therefore, we argue that it is imperative for students to know about prevailing health conditions, particularly new and contemporary issues, such as pandemics, which have recently plagued the world. The difficulties and deficits encountered owing to pandemics and epidemics, notably in the realm of education, have made it essential to incorporate information in scientific textbooks on how to handle different types of health crises. Hence, this study explores the extent to which middle school textbooks in Saudi Arabia include the health education requirements necessary to enable students to acquire healthy behaviors.

### 1.1. Problem

One of the key elements of the educational process is the inclusion of health literacy in science courses since its absence is detrimental to both education and society. Students will become more aware of these requirements if they are included in science courses, which could also strengthen their learning regarding health-related content [10]. The findings confirmed the importance of including health topics in school science curricula and covering them comprehensively. The implementation of the school health promotion program has led to positive changes in health behavior among children. To establish an effective health education system, the World Health Organization established the Committee of Experts on Health Education and Comprehensive Health Support through Schools in 1995 [8]. Studies [9] have recommended the incorporation of health education concepts into the school curriculum; this was realized to be the only way to spread awareness and reduce health risks among students.

There is a pressing need to incorporate health education into science curricula for middle school students as they are in the intermediate stage, which is considered crucial to their operational development. Health education aims to increase community health awareness by educating middle school students about health, helping them to develop healthy attitudes and habits, and involving them in health-promotion activities that, in turn, cause positive changes in their behavior. Hahn and Truman [1] indicated that the development of a student's capacity to acquire sound concepts, values, and attitudes depends

heavily on this type of education. Therefore, an evaluation of the intermediate-stage science curriculum regarding the inclusion of health literacy requirements is urgently required.

Thus, the purpose of this study is to understand emerging and modern health issues, such as pandemics, and examine middle school science textbooks in the Kingdom of Saudi Arabia to ascertain the degree to which these issues are addressed. In conclusion, this study aims to determine the extent to which health education requirements are incorporated into the content of intermediate-level science textbooks.

### 1.2. Research Questions

- What is the extent of the inclusion of health requirements in the content of middle school science textbooks in Saudi Arabia?
- Are there statistically significant differences in the extent of the inclusion of health education requirements in middle school science textbooks according to grade level?

## 2. Method

This study used a descriptive methodology and the content-analysis method, which is a research technique used to produce meaningful and organized quantitative descriptions of content and enables the explanation, collection, and analysis of data relevant to a study's goals to draw conclusions.

### 2.1. Sample

Six science books of the 2022 edition, to be taught in the first, second, and third middle school grades, representing the research sample (one book per semester for each grade) were analyzed. The characteristics of the study sample are presented in Table 1.

**Table 1.** Characteristics of the analyzed intermediate science textbooks.

| Grade | Term | # Pages | # Units | # Chapters | # Lessons |
|---|---|---|---|---|---|
| First | 1 | 218 | 3 | 6 | 14 |
| | 2 | 227 | 3 | 7 | 14 |
| Second | 1 | 214 | 3 | 6 | 13 |
| | 2 | 212 | 3 | 6 | 14 |
| Third | 1 | 280 | 4 | 8 | 19 |
| | 2 | 139 | 2 | 4 | 9 |

### 2.2. Instrument

A content-analysis card was used to explore the extent to which the content of the science textbooks met the health education requirements. To construct the textbook-analysis card, we searched and carefully read the relevant theoretical literature and prior studies (Nutbeam; Ullah, Naz; Ahmad; Humayun, and Ashraf, 2020; Knise, Rupprich, Wunram, Bremer, and Desaive) [11–13]. The initial list of health education requirements included the following themes: the link between exercise and mental health, physical activity (4 indicators), the mental health of the body (4 indicators), health education requirements related to personal health (5 indicators), health education related to community health (3 indicators), and health education related to epidemic prevention measures (3 indicators). The fourth requirement was health education related to epidemic prevention measures (three indicators). Healthy culture related to nutritional health (4 indicators) was the seventh requirement, the third requirement (5 indicators), and the eighth requirement (4 indicators). Finally, the ninth requirement regarding the genetic disease and reproductive health policies comprised 6 indicators.

### 2.3. Validity and Reliability

To verify the degree to which paragraphs (indicators) belong to a particular health requirement, as well as the accuracy and clarity of language (indicators) formulation, the study tool was initially presented to ten arbitrators with specialization and experience in



the fields of science, general curriculum development, and methods of teaching. Indicators were added, changed, or removed based on the arbitrators' observations. Consequently, the following four indicators were added:

Indicator 1: complied with the first requirement by describing chronic diseases and treatment options;

Indicator 2: the fourth requirement was expanded to include knowledge of the characteristics of infectious organisms (bacteria, fungi, and viruses);

Indicator 3: the sixth requirement was expanded to include awareness of diseases linked to obesity;

Indicator 4: the seventh requirement was expanded to additionally explain the importance of genetic tests before marriage.

Based on the arbitrators' remarks, the fourth indicator, "preserving environmental diversity", and the fifth, "preserving animal diversity", were merged into a single indicator of, "calls for the preservation of animal and plant environmental diversity", under the fifth requirement. The verbiage of some indicators were modified, including changing the third indicator in the first requirement, "awareness of diseases or conditions of health conditions of the elderly", to "awareness of diseases and health conditions of the elderly", changing the second indicator of the fourth requirement, "knowledge enhancement of community prevention methods", to "enhanced awareness of community prevention methods", and changing the first indicator of the fifth requirement, "the importance of ensuring adequate housing in a healthy way", to "enhanced the importance of adapting to a healthy home". A clarification was added to the second indicator of the sixth requirement, "awareness of diseases resulting from malnutrition, such as anemia", by adding more illustrative examples: "It raises awareness of diseases resulting from malnutrition, such as anemia, osteoporosis, and scurvy". After making modifications according to the opinions of the arbitrators, the final list of analysis cards comprised 27 indicators classified into 7 requirements as follows:

- The first requirement: health education related to physical activity and the mental health of the body (4 indicators);
- The second requirement: health education related to personal health (5 indicators);
- The third requirement: health education related to community health (3 indicators);
- The fourth requirement: health education related to epidemic prevention measures (4 indicators);
- The fifth requirement: healthy culture related to environmental health (4 indicators);
- The sixth requirement: healthy culture related to nutritional health (4 indicators);
- The seventh requirement: health education related to genetic diseases and reproductive health (3 indicators).

*2.4. Preparing the Data-Collection Guide*

The card was prepared in its final form after the validity of the tool had been confirmed in accordance with the arbitration notes. A guide for data collection was created and divided into the following four sections:

Section 1: the research objectives and questions were described in the introduction, which is the first section;

Section 2: the main categories and subcategories of analysis were explained in this section, along with what constitutes an analysis unit and what does not;

Section 3: this section explained the steps taken by the analyst to analyze books using the categories and units of analysis listed in the second section;

Section 4: a form was created to record the indicators that appeared in any analysis unit, making it easier to monitor and record data.

2.4.1. Validity of Data-Collection Evidence

To guarantee the appropriateness of the analysis categories, units, and techniques, the data-collection guide was given to four arbiters with expertise and experience in the fields of curriculum development and methods of teaching science. After examining the arbitration reports, the arbitrators approved the appropriateness of the analysis categories and units, as well as the accuracy and suitability of the analytical process.

2.4.2. Stability of the Analysis Tool

Two raters with the same area of expertise analyzed the content, wherein the following steps were conducted to compute stability:

1.  Five units were analyzed collectively to ensure that there was a common understanding of the categories, units, and procedures of analysis;
2.  The first unit of the first-grade middle school textbook from the first semester, was randomly selected to represent the stability sample. The unit included 6 lessons and 8 activities with 14 units of analysis; each rater analyzed the data independently. Table 2 shows the number of units of agreement and the percentage of stability among the raters. The percentage stability of the analysis among the raters ranged from 96.5% to 100%, with a value of 99.2% for the whole tool. These results indicate a high percentage of stability, demonstrating the stability of the tool and the appropriateness of its use in the analysis of similar content by different researchers.

**Table 2.** The stability of the analysis tool according to different raters.

| Requirements | No of Indicators | No. of Agreements | No. of Differences | % Agreement |
|---|---|---|---|---|
| Health education related to the physical activity of the body | 56 | 54 | 2 | 96.5 |
| Health education related to personal and psychological health | 70 | 70 | 0 | 100 |
| Health education related to community health | 42 | 42 | 0 | 100 |
| Health education related to epidemic prevention measures | 56 | 56 | 0 | 100 |
| Health education related to environmental health | 56 | 55 | 1 | 98.2 |
| Health education related to nutritional health | 56 | 56 | 0 | 100 |
| Health education related to genetic diseases and reproductive health | 42 | 42 | 0 | 100 |
| Total | 378 | 375 | 3 | 99.2 |

In analyzing middle school science textbooks, the researchers followed a precise analytical description according to the following procedural steps:

3.  Determining the analysis categories: these were the health education requirements and their indicators that formed the analysis phrases of the analysis card, which included 7 requirements and 27 indicators;
4.  Determining the unit of analysis: the units of analysis of the middle school science textbooks were classified into two types according to the content:

    -   The entire lesson, including the associated objectives, content paragraphs, review vocabulary, new vocabulary, main headings, subheadings, figures, pictures, tables, charts, and content boxes;
    -   Activities, including scientific experiments, problem-solving laboratories, real-life investigations, and the application of science and technology in society.

5.  The method of analyzing the content and judging whether the analysis of each unit included the indicator was as follows:

- The researchers distinguished between what is classified as a unit of analysis and what cannot be classified as a unit of analysis;
- The researchers started by defining the units of analysis in the textbook, such that they are numbered sequentially starting from number 1 until the completion of the book, according to the rules for choosing the unit of analysis;
- Each unit of analysis was categorized into two sections: the whole lesson and the associated content, pictures, tables, forms, and activities;
- All indicators were tracked in each unit of analysis, such that if the analyst found a phrase corresponding to any indicator of the tool, they recorded it by marking it with the symbol (1); if they did not, it was recorded with the symbol (0);
- If the analyst found more than one indicator in the same unit of analysis, they recorded all indicators; if the same indicator was found more than once in the same unit of analysis, the indicator was recorded only once;
- When an indicator was recorded in a particular unit of analysis and the same indicator was found in the next unit, the indicator was recorded again;
- The extent of the inclusion of each health education requirement (main analysis categories) was based on the frequencies and percentages of the indicators of the requirement (subcategories);
- The extent of the inclusion of the requirements was calculated based on the number of analysis units (the sum of the number of analysis units in the first, second, and intermediate third grades).

The calculation of the inclusion ratios was as follows:

The percentage of health education requirements included was calculated based on the number of analysis units. The number of middle school textbooks reached 187 units of analysis, with 66 units of analysis in the first grade, 60 units of analysis in the second grade, and 61 units of analysis in the third grade.

The inclusion percentages of the indicators were calculated via the following equations:

$$\text{The percentage} = \text{(the number of the frequencies in)} \div \text{((the number of total analysis units in the textbook} \times \text{the number of the indicators)} \times (100))$$

The average number of occurrences of the requirement appearing was calculated as:

$$\text{The percentage of inclusion of any indicator} = \text{(the number of occurrences of the indicator in the class)} \div \text{((the number of total units of analysis in the book)} \times (100))$$

The inclusion percentages of the indicators were calculated using the following equation:

$$\text{The percentage of including the first requirement in the first intermediate grade} = \text{(the number of repetitions of the requirement appearing in the first intermediate grade)} \div \text{((the number of analysis units in the two science textbooks for the first intermediate grade)} \times \text{the number of indicators of the first requirement} \times (100)).$$

Example 1 shows the calculation of the percentage inclusion of the first requirement in the first intermediate grade:
The number of analysis units in the intermediate first book = 66 units of analysis;
The first requirement appeared four times;
The number of indicators of the first requirement = 4.
Then:

$$\text{The percentage of including the first requirement in the first intermediate grade} = 4 \div (66 \times 4) \times 100 = 4 \div 264 = 1.52\%$$

Example 2 shows the calculation of the ratio of any indicator that "supports the practice of physical activity" in the intermediate first-grade textbook:

Percentage of including "supports the practice of physical activity" in the intermediate first-grade book = (the number of repetitions of the indicator appearing in the intermediate first grade) ÷ ((the number of total analysis units in the intermediate first-grade books) × (100)

The indicator supporting the practice of physical activity appeared four times in the intermediate first-grade textbook; therefore, the inclusion rate was calculated as follows:

The percentage of including "supports the practice of physical activity" in the intermediate first grade book = 4 ÷ (66) × 100 = 6.06%.

### 3. Findings

This section presents the answers to the research questions by extracting the results of the application of the research tool, statistically analyzing and processing the data, and interpreting and discussing them. A summary of the results for the three classes is presented, followed by the results for each class.

Q1: What is the extent of the inclusion of health education requirements in science books for the intermediate stage?

As seen in Table 3, all the health education requirements were inadequately included in the science books for the intermediate stage, wherein the inclusion rates ranged between 5.0% and 1.3% and the inclusion rate, in general, reached only 2.8%. The results show that the fifth requirement, "*health education related to environmental health*", ranked first, with an inclusion rate of 5.0%. The second requirement, "*Health education related to personal and psychological health*", came in second place, with an inclusion rate of 3.5%. The first requirement, "*health education related to the physical activity of the body*", ranked third, with an inclusion rate of 2.7%. In fourth place was the sixth requirement, "*health education related to nutritional health*", with an inclusion rate of 2.5%. In fifth place was the fourth requirement, "health education *related to epidemic prevention measures*", with an inclusion percentage of 2.1%; the third requirement, "*health education related to community health*", was sixth, with an inclusion rate of 1.4%. Finally, "*Health education related to genetic diseases and reproductive health*" had an inclusion rate of (1.3%).

**Table 3.** Frequency percentage of the extent of the inclusion of health education requirements in science books for the intermediate stage.

| Requirements | No. of Indicators | First Grade (N = 66) f | % | Second Grade (N = 60) f | % | Third Grade (N = 61) f | % | Total (N = 187) f | % | Rank |
|---|---|---|---|---|---|---|---|---|---|---|
| Health education related to the physical activity of the body | 4 | 4 | 1.52 | 12 | 5.0 | 4 | 1.7 | 20 | 2.7 | 3 |
| Health education related to personal and psychological health | 5 | 9 | 2.73 | 17 | 5.7 | 6 | 2.0 | 32 | 3.5 | 2 |
| Health education related to community health | 3 | 2 | 1.0 | 2 | 1.1 | 4 | 2.2 | 8 | 1.4 | 6 |
| Health education related to epidemic prevention measures | 4 | 2 | 0.75 | 10 | 4.2 | 3 | 1.2 | 15 | 2.1 | 5 |
| Health education related to environmental health | 4 | 17 | 6.44 | 14 | 5.8 | 7 | 2.9 | 38 | 5.0 | 1 |
| Health education related to nutritional health | 4 | 2 | 0.75 | 14 | 5.8 | 2 | 0.80 | 18 | 2.5 | 4 |
| Health education related to genetic diseases and reproductive health | 3 | 0 | 0 | 5 | 2.8 | 2 | 1.1 | 7 | 1.3 | 7 |
| All health requirements | 27 | 36 | 2.02 | 74 | 4.57 | 28 | 1.7 | 138 | 2.8 | |

The findings show that science books for the third intermediate grade were the least inclusive of health education requirements and the second intermediate grade textbooks were the most inclusive. Moreover, the fifth requirement, "*health education related to environmental health*", was the most-included requirement in science textbooks for the first and third middle school grades; whereas, the second requirement, "*health education related to personal and psychological health*", was the most-included requirement for the second middle school grade. The science textbook for the second middle school grade included

the seventh requirement of "health education related to genetic diseases and reproductive health" at a higher rate than the first and third grades; in fact, the seventh requirement was not included in the science books for the first grade.

As Table 3 shows, all health education requirements were included in the science textbooks of all grades, except for the seventh requirement, which was not included in the science textbook for the first intermediate grade. However, the inclusion rates ranged between 6.44% and 0.75%, which is weak.

Q2: What is the statistically significant difference in the extent of the inclusion of health education requirements according to grade level?

To answer this question, frequencies, percentages, chi-square values ($\chi^2$), and their significances were calculated for each health education requirement. The results in Table 4 show that the science textbook for the second intermediate grade is the highest ranked in terms of including the first, second, fourth, sixth, and seventh requirements compared to the first and third intermediate grades, with a statistically significant difference; no statistically significant differences appeared between the three grades regarding the extent of the inclusion of the third and the fifth requirements.

**Table 4.** Chi-square ($\chi^2$) of the differences for the inclusion of health education requirements, according to grade.

| | Requirements | Grade | Units Included f | % | $\chi^2$ | $p$ |
|---|---|---|---|---|---|---|
| 1. | Health education related to the physical activity of the body | First<br>Second<br>Third | 4<br>12<br>4 | 1.52<br>5.0<br>1.7 | 8.017 | 0.018 |
| 2. | Health education related to personal and psychological health | First<br>Second<br>Third | 9<br>17<br>6 | 2.73<br>5.7<br>2.0 | 8.165 | 0.017 |
| 3. | Health education related to community health | First<br>Second<br>Third | 2<br>2<br>4 | 1.0<br>1.1<br>2.2 | 1.156 | 0.561 |
| 4. | Health education related to epidemic prevention measures | First<br>Second<br>Third | 2<br>10<br>3 | 0.75<br>4.2<br>1.2 | 6.185 | 0.045 |
| 5. | Health education related to environmental health | First<br>Second<br>Third | 17<br>14<br>7 | 6.44<br>5.8<br>2.9 | 4.489 | 0.106 |
| 6. | Health education related to nutritional health | First<br>Second<br>Third | 2<br>14<br>2 | 0.75<br>5.8<br>0.80 | 14.031 | 0.001 |
| 7. | Health education related to genetic diseases and reproductive health | First<br>Second<br>Third | 0<br>5<br>2 | 0<br>2.8<br>1.1 | 6.111 | 0.047 |

## 4. Discussion

Regarding the answer to the first question, which was centered on the extent of the inclusion of health education requirements in science books for the intermediate stage; the results indicated that all health education requirements were poorly included in the science textbooks of all grades, except for the seventh requirement, which was not included in the science textbook for the first intermediate grade.

These results were consistent with those of several others conducted inside and outside of the Kingdom of Saudi Arabia. For example, Begoray, Higgins, and MacDonald [3] showed that the content of science textbooks for the first and second intermediate grades lacks health and environmental concepts, trends, and practices. This is also consistent with literature [7] reporting that the content of the science curriculum for the intermediate first grade in the Kingdom of Saudi Arabia does not adequately consider health education standards and [14] the inclusion of elements of interest in health and safety in science books in primary schools in the Kingdom of Saudi Arabia for the academic year 2009 was weaker than in previous periods.

The results of this study are also in agreement with a number of regional studies, such as a study conducted by Kruk et al. [15] which showed that the degree of the inclusion of

health concepts in science books obtained the highest rates; whereas, the trends obtained the lowest. In the study of Sibley [16], it was concluded that the extent of the inclusion of preventive education standards in science textbooks for the fourth grade in the fields of healthy nutrition, disease prevention, environmental protection, protection of the human body, safety, and security was not at the required level compared with the considered standard of 70%. Globally, the results of this study agree with those of Omar et al., (2015) [17], who showed a deficit in health education regarding hand washing and personal hygiene in science curricula in Pakistan, and Hussain, Alamgir, and Shahzad (2015) [18], who stated that the basics of health education are insufficiently included in the curricula at the primary stage in Pakistan.

On the other hand, the results of this study disagree with that of [12], who indicated that the percentage of health fields covered by general science textbooks for the fifth, sixth, and seventh grades of the basic stage in Palestine reached 72.7% and the percentage of environmental fields covered reached 66.7%. It also disagreed with a study by Al-Shehhi, Emam, Al-Otaiba, Ibrahim, and Al-Mehrizi [19], which reported that the effectiveness of promoting the healthy behavior of the -science curriculum developed for the first grade in Jordan was rated moderately in most evaluation indicators by supervisors and teachers.

Regarding the findings of the chi-square ($\chi^2$) test, which related to the second question, it was found that the science textbook for the second intermediate grade is ranked the highest in terms of including the first, second, fourth, sixth, and seventh requirements compared to the first and third intermediate grades, with a statistically significant difference; no statistically significant differences appeared between the three grades in terms of the extent of the inclusion of the third and fifth requirements.

These results proved that even though there were statistically significant differences among grades regarding the inclusion of some requirements, the extent of the inclusion of each of those requirements was very poor. Thus, it is very important to develop science textbooks with the inclusion of health education literacy in mind.

Health is something that many people want but struggle to achieve and maintain. That is why it is so important for people to develop their information and literacy skills in the health field. "Health literacy", which is defined as the capacity to read, write, and speak effectively about health [8], is a measure of a person's capacity to comprehend and communicate health-related information. Health education must be emphasized in science curricula in light of the emergence of new health problems as a means of protecting societies against contracting era-specific diseases and increasing community understanding of health issues that will guarantee the avoidance of diseases. In light of recent crises and disasters caused by pandemics and epidemics, such as SARS, avian flu, Ebola, and dengue fever, the significance of health-promotion methods and sickness-prevention methods has been emphasized by experts in the field of health education and by science educators as well [9].

Health sustainability is supported through health prevention and health education. To ensure future sustainability, students must learn how to stay healthy. Today's schools must help students overcome their ignorance and correct their misperceptions about the world. It must help them adopt healthier, more sustainable habits for themselves and future generations. A science curriculum reform would ensure students' longevity and wellness. Integrating health education values and standards into science textbooks promotes healthy and sustainable growth.

## 5. Conclusions

This study aimed to identify the extent of the inclusion of health education requirements in the content of middle school science textbooks. The results revealed that all health education requirements were included in science textbooks with very weak inclusion rates in all grades. This study is significant from a theoretical and practical standpoint. Theoretically, it addresses a crucial topic pertaining to health education, particularly in light of the coronavirus pandemic and the breadth of its influence across all facets of life. The findings of this analysis may shed light on the requirements of middle school health education that

should be incorporated into science textbooks. Specialists should create curricula in general and science courses for this stage in particular to meet the health education requirements for all students. Practically, it might help decide the extent to which health education requirements should be incorporated into middle school science textbooks in the Kingdom of Saudi Arabia or any other country. This study also offers a list of the content that was used to generate the findings and suggestions. Finally, this might assist researchers in future investigations of scientific curricula and it might have day-to-day applications. In light of this study's findings, it is important to consider the inclusion of health education related to hereditary diseases and reproductive health in science textbooks to develop students' awareness to avoid perpetuating genetic disorders and stress the importance of genetic screening prior to marriage. In addition, an awareness of the characteristics of diseases borne by bacteria, fungi, and viruses is required. This can be achieved by including first-aid techniques and ways to combat harmful societal habits in science textbooks at the intermediate stage as they are essential to community health. It is also necessary to include content that promotes awareness of healthy habits in general, describes the negative effects of unhealthy habits, supports mental health, and conveys the significance of using health tools in spreading awareness and disease-prevention understandings, locally and globally. This would strengthen cultures related to personal and psychological health. Further research, including an analytical study, is suggested to determine the extent to which secondary school textbooks for chemistry, physics, biology, and geosciences include health education requirements. Further study is required to compare the extent to which these requirements are included in various courses and grades and examine the awareness of science teachers regarding the needs of health students and communities in terms of health education.

**Author Contributions:** Conceptualization, A.H.A. and S.A.A.; Methodology, S.A.A.; Validation, A.H.A. and S.A.A. Formal Analysis, A.H.A. and S.A.A.; Investigation, A.H.A. and S.A.A.; Resources, S.A.A.; Data Curation, A.H.A.; Writing—Original Draft Preparation, S.A.A.; Writing—Review and Editing, A.H.A.; Visualization, A.H.A.; Supervision, A.H.A.; Project Administration, A.H.A.; Funding Acquisition, A.H.A. All of the authors have sufficiently contributed to this study and agree with its results and conclusions. All authors have read and agreed to the published version of the manuscript.

**Funding:** This research was funded by the Deanship for Research and Innovation, "Ministry of Education" in Saudi Arabia (IFKSUOR3-257).

**Institutional Review Board Statement:** Not applicable.

**Informed Consent Statement:** Not applicable.

**Data Availability Statement:** All data are included in the results.

**Acknowledgments:** The authors extend their appreciation to the Deanship for Research and Innovation, "Ministry of Education" in Saudi Arabia, for funding this research (IFKSUOR3-257).

**Conflicts of Interest:** The authors declare no conflict of interest.

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
