# Peer review of "The Extent of Incorporating Health Education Requirements in Middle School Science Textbooks"

_sustainability, doi:10.3390/su151411005_

Round 1
Reviewer 1 Report
The author (s) tried to discover the level of the inclusion of health education requirements in the content of middle school science textbooks. Although the idea is good, the innovativeness is very low. The statistical procedure and generalizability is also very poor. The overall manuscript should be improved a lot. Thank you.
Need to copy-edit the manuscript
Author Response
We wish to re-submit the manuscript titled “The Extent of Incorporating Health Education Requirements in Middle School Science Textbooks.” The manuscript ID is sustainability-2364687.
We thank you and the reviewers for your thoughtful suggestions and insights. The manuscript has benefited from these insightful suggestions. I look forward to working with you and the reviewers to move this manuscript closer to publication in the Sustainability.
The manuscript has been rechecked and the necessary changes have been made in accordance with the reviewers’ suggestions. These changes have been marked using the ‘Track changes’ function on Microsoft word. The responses to all comments have been prepared and presented below.
NOTE:
It should be noted that some additional corrections have been made to the CLEAR copy, which may not appear in the TRACKED copy. Therefore, the clear copy should be considered.
Attached are; 1) Tracked copy of the revised manuscript and. 2) a CLEAR copy of the revised manuscript
Thank you for your consideration. I look forward to hearing from you.
Sincerely,
Abdulwali H Aldahmash
Professor of Science Education, Curriculum and Instruction Dept., College of education, & Excellent Centre for Science and Mathematics Education, King Saud University, Saudi Arabia
[00966557671512]
aaldahmash@ksu.edu.sa

Reviewer 2 Report
Dear authors,
Congratulations for the topic covered. It is an interesting one and you have analyzed it at a proper scientific level.
My recommendation is to (re)read the journal recommendations regarding how to cite references in the text but also in the final list of recommendations as well as how to organize the manuscript.
Best regards
Author Response

(The authors gave the same response as above.)

Reviewer 3 Report
The research address contemporary questions and the goal of the research is significant to both filling the academic gap in science education and textbook content for grade students.
Then researcher utilized the content analysis card... to determine the extent to which the content of the targeted textbooks met the needs for health education culture".
The academic rigor demonstrated by the author is of acceptable standard, including the research design, data analysis process, validation of results and contribution to a relevant field of knowledge (science education).
Some minor grammar issues need to be fixed in the paper. Overall quality of presentation and language use was good.
Author Response

(The authors gave the same response as above.)

Reviewer 4 Report
This article deals with the exciting topic of incorporating "Health Education Requirements" in schools. However, the text needs some improvement.
- The manuscript has serious writing problems, misspelt words, incorrect punctuation marks, incorrect use of parentheses, and double spaces. Lines 12, 63, 65, 84, 93, 97, 100, 122, 123, 199, 217, 249, 251-253, 292, 295, 317, 318, 320, 323, 327, 329, 330, 332, 334, 351, among others, should be revised.
- The article's objective appears in the Abstract and the Conclusions but not in the Introduction. The Introduction says that "the purpose of this study is to understand health issues, particularly emerging and modern issues, such as pandemics that affected the entire world, and then examine student science textbooks for the intermediate stage in the Kingdom of Saudi 99 Arabia".
- Although it is a quantitative study, it does not present hypotheses.
- The methodology needs to be more precise regarding the method, techniques, research instruments, data analysis, etc.
- The relationship between the numbers of the indicators (1, 2, 3, 4) and the "Health Education Requirements" is unclear. The numbers do not correspond, and it is not easy to understand.
- I suggest reducing the lines from 213 to 248. It is optional to give so much detail.
- A more in-depth discussion of the results needs to be included.
- The short discussion appears in the Results section; there is no Discussion section.
- Why is the first sentence of Table 4 in bold type?
- The article should change how it presents the ideas to be better understood. I suggest putting what the "Health Education Requirements" are and, from there, pointing out what has been incorporated and what is missing.
- Look for more attractive ways of presenting the data (comparison between advances and absences, graphs or figures, etc.).
it is not easy to read
Author Response
Dear reviewer:
We wish to re-submit the manuscript titled “The Extent of Incorporating Health Education Requirements in Middle School Science Textbooks.” The manuscript ID is sustainability-2364687.
We thank you and the reviewers for your thoughtful suggestions and insights. The manuscript has benefited from these insightful suggestions. I look forward to working with you and the reviewers to move this manuscript closer to publication in the Sustainability.
The manuscript has been rechecked and the necessary changes have been made in accordance with the reviewers’ suggestions. These changes have been marked using the ‘Track changes’ function on Microsoft word. The responses to all comments have been prepared and presented below.
NOTE:
It should be noted that some additional corrections have been made to the CLEAR copy, which may not appear in the TRACKED copy. Therefore, the clear copy should be considered.
Attached are; 1) Tracked copy of the revised manuscript and. 2) a CLEAR copy of the revised manuscript
Thank you for your consideration. I look forward to hearing from you.
Sincerely,
Abdulwali H Aldahmash
Professor of Science Education, Curriculum and Instruction Dept., College of education, & Excellent Centre for Science and Mathematics Education, King Saud University, Saudi Arabia
[00966557671512]
aaldahmash@ksu.edu.sa

Reviewer 5 Report
The authors tried to expolre the level of the inclusion of health education requirements in the content of middle school science textbooks and found all health education requirements were included in those science textbooks with a very weak inclusion at all grades. The theme has very important theoretical and practical implication. However, I would advise the authors to revised thses points prior to publishing the article in its final form.
1. There are problems with the format of the reference. For example, "(Darling-Hammond & Hyler, 2020) stressed that the school's role currently extends beyond merely preparing students in a thorough and balanced manner in terms of their academic … (p1) "should be changed to “Darling-Hammond & Hyler (2020) stressed that the school's role currently extends beyond merely preparing students in a thorough and balanced manner in terms of their academic …(p1)”
2. The author should provide the meaning of the p-value in Chi-square analysis.
3. The authors did not adopt strict three-line form in Table 3 and Table 4, so it is suggested that the author modify them.
Author Response

(The authors gave the same response as above.)

Round 2
Reviewer 4 Report
I hope that my suggestions have helped to improve the manuscript.